# Friction Behavior of Pre-Damaged Wet-Running Multi-Plate Clutches in an Endurance Test

**Thomas Schneider \*** , **Katharina Voelkel** , **Hermann Pflaum and Karsten Stahl**

Gear Research Centre (FZG), Technical University of Munich, 85748 Garching, Germany;
voelkel@fzg.mw.tum.de (K.V.); pflaum@fzg.mw.tum.de (H.P.); stahl@fzg.mw.tum.de (K.S.)

**\*** Correspondence: schneider@fzg.mw.tum.de

**Abstract:** Wet-running multi-plate clutches should be prevented from failing due to the often safety-relevant functions they fulfill in the drive train. In addition to long-term damage, spontaneous damage is of particular relevance for failures. This paper focuses on the influence of spontaneous damage on frictional behavior in the later life cycle. The aim of the experimental investigations is to initially cause spontaneous damage in wet-running multi-plate clutches with sintered friction linings. For this purpose, three clutches are first pre-damaged in stage tests with different intensities, so that the first spontaneous damage (local discoloration, sinter transfer) occurs. In the second step, an endurance test is carried out with the pre-damaged clutch packs and a non-pre-damaged reference clutch. The friction behavior of the clutches during the endurance test is compared and evaluated. It shows that local discoloration and sinter transfer are no longer visible after the endurance tests. At the beginning of the endurance test. the values of coefficient of friction are higher over the entire speed range of the heavily pre-damaged clutches than with the slightly pre-damaged clutch and the non-pre-damaged reference clutch. At the end of the endurance test, it can be observed that the greater the pre-damage to the clutches is. the greater the coefficient of friction increases with decreasing sliding speed.

**Keywords:** friction behavior; wet friction clutch; damage behavior; spontaneous damage; experimental tests; continuous shifting tests

## 1. Introduction

Wet multi-plate clutches and brakes are widely used machine elements in drive technology. They belong to the group of shiftable friction clutches and can be engaged and disengaged under load. They are used in automobiles as shift elements in automatic transmissions and non-permanent all-wheel drives, as input elements in dual clutch transmissions and as differential locks. In industrial applications, they are used both as shift clutches, as slip clutches to protect against overload or vibration and as wet brakes of off-road vehicles.

Rising performance requirements and increasing cost pressure require continuous improvement in design and operational reliability. In view of the often safety-relevant use of multi-plate clutches, it must be ensured that they operate reliably and without damage under high loads or in individual overload situations. For this purpose. the load capacity of the clutch must be known. This requires knowledge of the damage mechanisms and influencing variables. Damage to wet multi-plate clutches is caused by high mechanical and/or thermal loads and can be divided into spontaneous and long-term damage, depending on the time course [1]. In this paper the influence of spontaneous damage on the shifting behavior of clutches during extended use will be investigated.

Long-term damage occurs over a long period of time—sometimes tens of thousands of engagements. Typical long-term damage is continuous wear or changes in frictional behavior over the operating period.

Acuner et al. [2] carried out endurance tests and described a clogging of the porosity of the lining under high thermal load for organic carbon friction linings. This smoothing of the friction lining by oil-cracking products leads to a reduction of the dynamic friction coefficient during the shifting process. The experimental investigations of Matsumoto [3] and Nyman et al. [4] also showed a correlation between lining porosity, friction coefficient and service life for wet-running multi-plate clutches. It was shown that a porous material is more durable in terms of thermal properties. In addition to porosity, friction surface temperature also has a major influence on the lifetime behavior of clutches [5]. According to Li et al. [6], prolonged high temperatures can degrade the cellulose fibers of the paper friction material (coking). This reduces the strength of the lining and the wear rate may be increased. Hensel et al. [7] reported on reference tests consisting of step tests and long-term tests to check the friction plates for spontaneous and cumulative damage. The influence of the operating mode on the damage can be determined by temperature measurements based on the influences on the heat load of the clutch or brake. Characteristic values for the evaluation of the thermal stress are derived from these measurements that allow a correlation with the long-term damage (decrease in the average friction coefficient).

In contrast to long-term damage of clutches, spontaneous damage is caused by overload within a few shifts. Some of the most well-known forms of spontaneous damage include hot spots with organic friction linings or sinter transfer in sintered metal friction materials. Both types of damage are due to high local thermal loads [8]. Hämmerl et al. [9] described hot spots as consisting of irregular yellow-brown to blueish discoloration on steel plates. Anderson and Knapp [10] and Fairbank et al. [11] distinguished two types of hot spots. In the first type, only "cosmetic" surface changes can be observed, whereas in the second type, plastic deformations occur on the steel friction surfaces. In the second type, local formation of martensite occurs, which indicates very high temperatures of up to 900 °C. Duminy et al. [12,13] investigated the performance limits of wet-running multi-plate clutches with the friction combination steel–sintered bronze. They described the following types of damage: sinter transfer, sinter surface lubrication, clogging of the lining pores, cracking of the lubricant and wear. Pfleger et al. [14] confirmed these results and distinguished between light, medium and strong sinter transfer depending on the increase in friction value. According to this. the light sinter transfer correlates with friction coefficient peaks of up to 15%, whereas the strong sinter transfer is characterized by an increase of the friction coefficient by at least 30%. The medium sinter transfer moves between the two levels. According to Strebel and Schneider et al. [15,16] spontaneous damage to steel plates with sinter metal friction linings can also be divided into two classes. On the one hand, bluish to black discolorations of the steel plate can be recognized; on the other hand, a transfer of friction material onto the steel plate can occur.

Zagrodzki et al. [17] investigated the generation of hot spots on wet-running multi-plate clutches during a short-term shifting process with changing speed. According to this study, small geometric errors, such as the waviness of the plates, can effectively trigger the process of hot spot formation during short-term clutch operation at high initial sliding speed.

Barber [18] used the theory of thermoelastic instability (TEI) to explain the formation of hot spots, according to which unequal load distributions in the contact area lead to excessive heat input and thus to local temperature increases. This leads to varying degrees of thermal expansion, which reinforces the original local pressure unevenness. These pressure increases lead to local thermal overloads. With low wear. the system becomes unstable and hot spots are formed. This effect is self-reinforcing and can destroy the friction system within a few shifts. A comprehensive model for analyzing the thermoelastic instability in wet-running multi-plate clutches is presented in [19]. The analysis shows the important role of permeability and deformation of the friction material. The strong influence of the elastic modulus on the formation of hot spots was confirmed in experimental tests by Fairbank [11].

The effects of friction material properties on thermoelastic instability and the associated dominant deformation mode were investigated in detail in [20]. They create both analytical and FE models to determine the variation of the critical velocity of the dominant TEI mode. The effects of inconsistent contact conditions were affirmed by studies of Zhao et al. [21], Takezaki and Kubota et al. [22].

## 2. Materials and Methods

Photographs of the investigated friction plates with sintered metal friction lining, as well as the corresponding steel plates, are shown in Figure 1. The technical data is listed in Table 1. The steel plates are used as inner plates and the friction plates are used as outer ones.

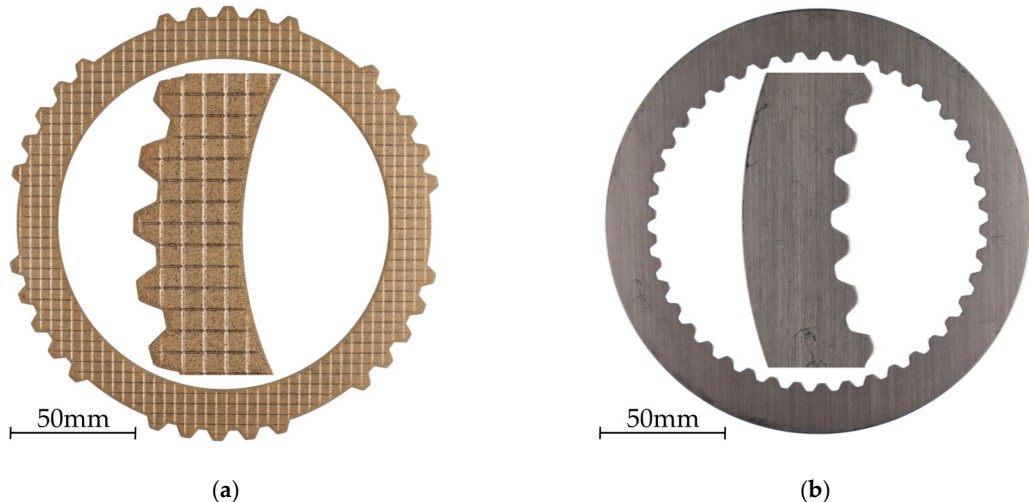

**(a)**                                                **(b)**

**Figure 1.** (**a**) Friction plate; (**b**) steel plate.

**Table 1.** Technical details of the plates.

| | |
|---|---|
| **Outer Diameter** | 164 mm |
| **Inner Diameter** | 132 mm |

To examine the load limits concerning spontaneous damage to multi-plate clutches, step tests were carried out on the ZF/FZG KLP-260 component test bench. A sketch of the test bench is shown in Figure 2. Sketch of the test bench ZF/FZG KLP-260 according to Meingaßner et al. [22].

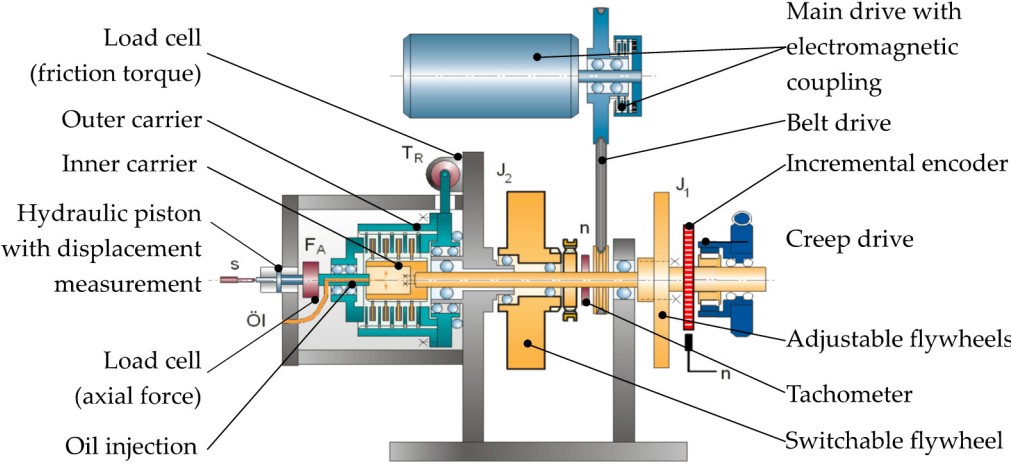

**Figure 2.** Sketch of the test bench ZF/FZG KLP-260 according to Meingaßner [23].

　　　The tests were carried out on the ZF/FZG KLP-260 clutch test bench. The test bench is a brake test bench with an outer carrier fixed to the housing, as well as a rotating inner carrier [23]. In the ZF/KLP-260 test bench, a complete clutch of the transmission is installed via corresponding drivers. The inner plates are arranged on the central shaft and connected to the flywheels (J1, J2). The contact pressure is applied to the outer plates, at which the friction torque of the clutch is also measured. The clutch is supplied centrally with cooling oil from the inside (defined cooling oil flow; oil temperature $\vartheta$oil thermostat-controlled heating and cooling). External lubrication from above is also possible.

　　　For each shifting test. the main shaft with internal driver and flywheel masses is accelerated by the speed-controlled main drive to the differential speed $\Delta$n. During the clutch test. the motor is separated from the test bench by an electromagnetic clutch. The shifting force is applied to the clutch by the hydraulic cylinder in a defined manner via the clutch-specific pressure ring. Force increase and force maximum are adjustable.

　　　In the slip operation, a defined axial force is applied to the clutch. The specified differential speed is set in the clutch via the creep drive. The slip speed is measured via a high-resolution incremental encoder.

　　　The mass moment of inertia J loading the clutch can be changed by flange-mounting discs of different mass moments of inertia J1 and connecting the basic mass J2. The technical data of the test bench are listed in Table 2.

**Table 2.** Technical data of the test bench ZF/KLP-260.

| Mass Moments of Inertia | $J_1 = 0.1...0.75$ | $kgm^2$ |
|---|---|---|
| | $J_2 = 1.0$ | $kgm^2$ |
| Outside disc diameter | $d = 75...260$ | mm |
| Max. friction torque | $T_{R, max} = 2000$ | Nm |
| Max. differential speed power shift mode | $\Delta n = 0...7000$ | $min^{-1}$ |
| Max. differential speed slip mode | $\Delta n = 140$ | $min^{-1}$ |
| Max. axial force | $F_{A, max} = 0...20$ | kN |
| Oil temperature | $\Theta_{oil} = 30...150$ | °C |

　　　Having described the test parts and the test bench. the method is explained in the following section. First. the clutches are run in to compensate for manufacturing tolerances and inhomogeneities. The specific loads (cf. Table 3) related to the friction surface while running in are based on the findings of Voelkel et al. [24].

**Table 3.** Loads in the run-in.

| Engagements | Pressure in $N/mm^2$ | Friction Work in $J/mm^2$ |
|---|---|---|
| 1000 | 0.5 | 0.5 |

　　　To investigate the friction behavior of pre-damaged multi-plate clutches in a continuous shifting test, these are subjected to prior damage in step tests. This type of pre-damage can also be considered as pre-conditioning for these investigations. The step tests, which are carried out on the ZF/FZG KLP-260 component test bench in braking operation, have already proven their validity in Hensel and Strebel et al. [1,25]. The initial speed of the circuit is increased from load level to load level in the step test. With ten engagements per load level. the number is sufficiently small to rule out long-term clutch changes and sufficiently large to detect any stochastic effects on spontaneous damage. After each load level. the clutch is inspected for any signs of damage and any changes are documented. The cycle time between two engagements is long enough (40 s) to ensure the recooling of the clutch system to oil injection temperature before each engagement. During running in and the step test. the specific cooling oil volume flow with respect to the clutch area for all clutches is 0.8 $\frac{mm^3}{mm^2 \; s}$. The load levels of the step test are listed in Table 4. All loads are normalized with respect to the clutch disc area.

**Table 4.** Load levels in the step test.

| Load Level | Pressure in N/mm² | Friction Work in J/mm² | Sliding Speed in m/s |
|:---:|:---:|:---:|:---:|
| 1 | 1.00 | 0.27 | 10.19 |
| 2 | 1.00 | 0.40 | 12.37 |
| 3 | 1.00 | 0.54 | 14.31 |
| 4 | 1.00 | 0.69 | 16.18 |
| 5 | 1.00 | 0.84 | 17.81 |
| 6 | 1.00 | 0.97 | 19.22 |
| 7 | 1.00 | 1.12 | 20.70 |
| 8 | 1.00 | 1.28 | 22.20 |
| 9 | 1.00 | 1.42 | 23.18 |
| 10 | 1.00 | 1.58 | 24.51 |

For a better overview. the loads are also shown graphically in the step-by-step test. Figure 3 illustrates the specific load levels of a step test with ten load levels. Each point represents a load level of the step test. The variation of the sliding speed $v_0$ in the individual load levels affects the friction work $q$ and peak friction power $\dot{q}_0$.

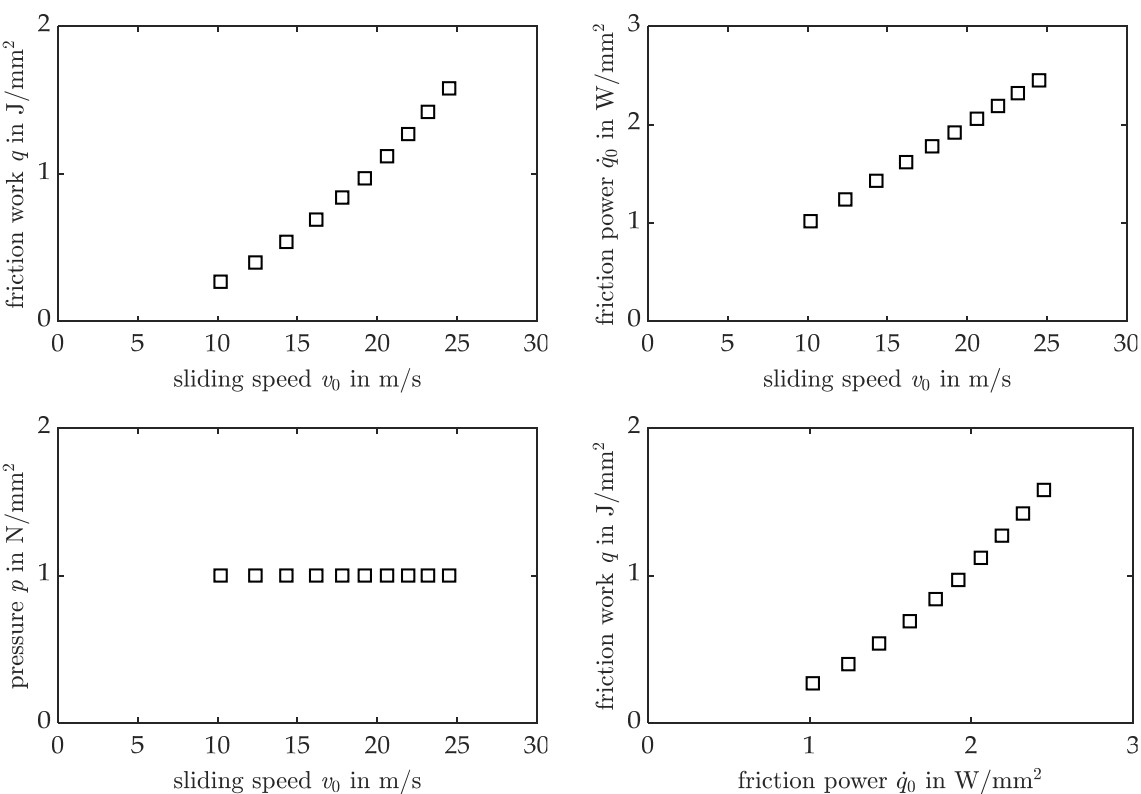

**Figure 3.** Load levels of the step test.

The pre-damage step test was followed by the endurance test with loads according to Table 5. During the endurance test. the specific cooling oil volume flow is 0.8 $\frac{mm^3}{mm^2\ s}$ and the cycle time 15 s.

**Table 5.** Load data in the endurance test.

| Engagements | Pressure in N/mm² | Friction Work in J/mm² |
|:---:|:---:|:---:|
| 10,000 | 0.5 | 0.5 |

To aid understanding. the experimental method is shown in a schematic diagram in Figure 4.

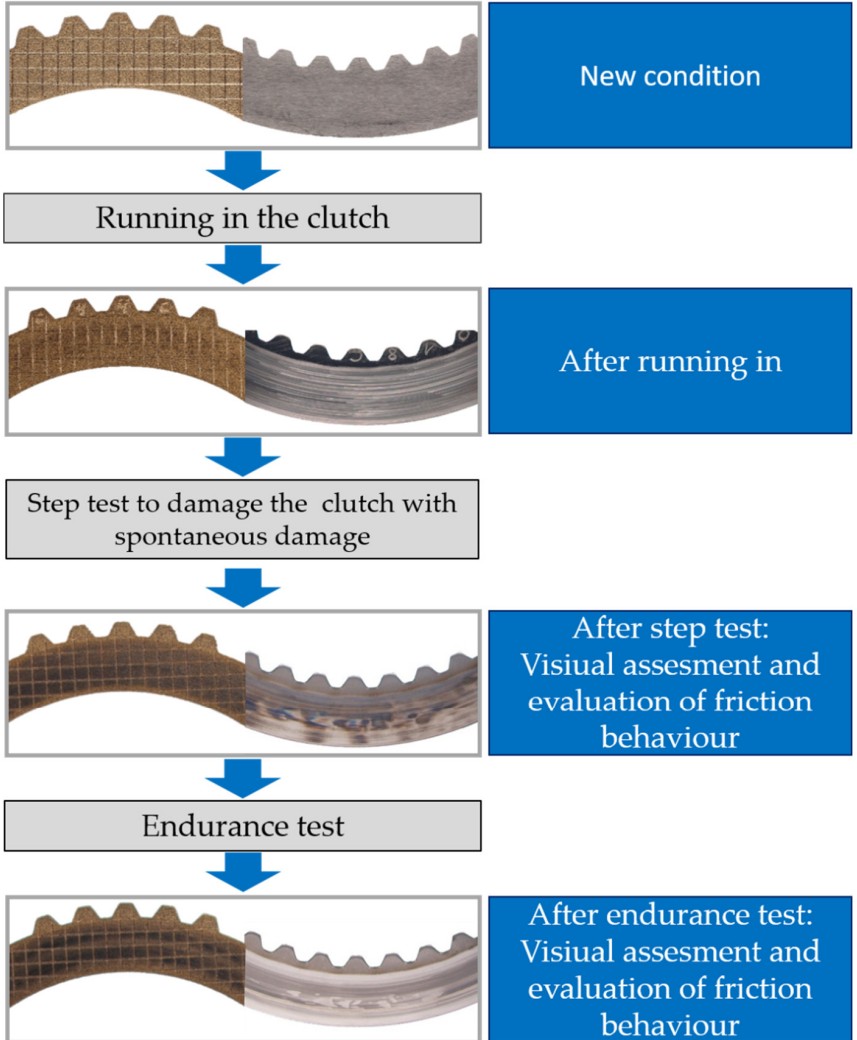

**Figure 4.** Experimental method.

## 3. Results

For the test LL-03. the step test was run up to load level 3 (cf. Table 4). These loads caused the first local discolorations on the steel plate. No fluctuations in the friction coefficient of a cycle were observed at load level 3. Figure 5 shows the friction coefficient curve for a shift at load level 3. The step test was followed by the endurance test.

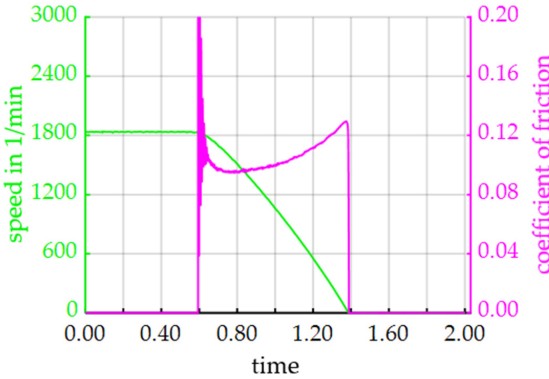

**Figure 5.** Friction coefficient curve for a shift at load level 3.

Figure 6a shows the steel and lining plate directly after the step test by pre-damaging the package. First discolorations on the steel plates can be observed. On the right-hand side (Figure 6b). the condition of the plates after the endurance test is documented. The photographs show that local discoloration on the steel plate is no longer visible after the endurance test. The friction plate shows a stronger discoloration after the 10,000 shifts of the endurance test.

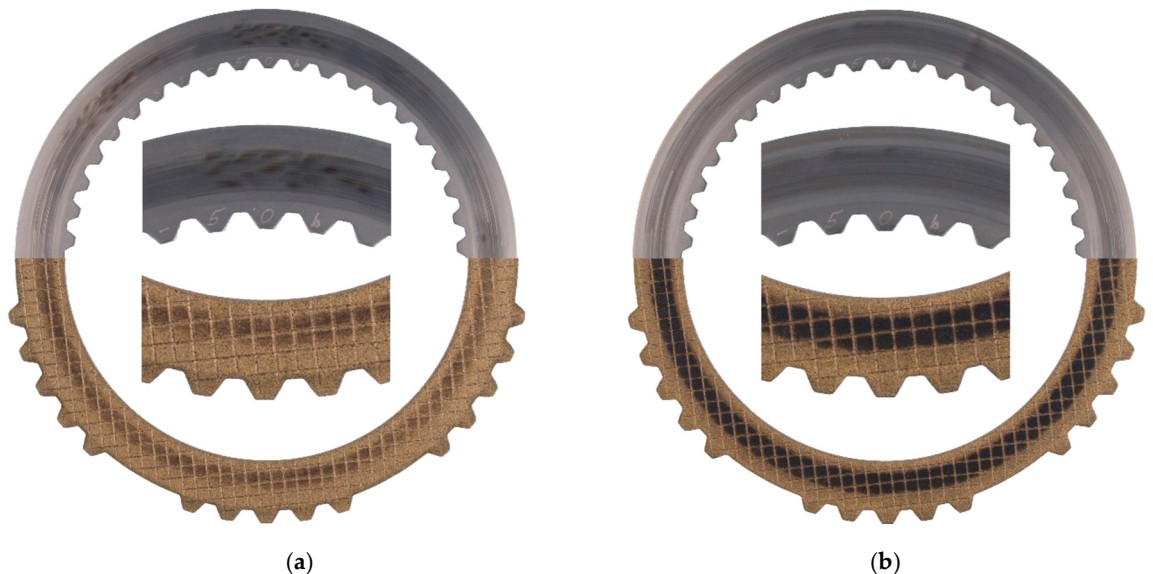

(**a**)                                                      (**b**)

**Figure 6.** (**a**) Photos of the pre-damaged plates after load level 3; (**b**) photos of the pre-damaged plates after the endurance test.

For the test LL-06. the step test was run up to load level 6 (cf. Table 4). The step test is carried out until local discoloration and a recognizable amount of sinter is transferred to the steel plate. At the same time, friction coefficient fluctuations as a result of seizing occur during the shifting process (Figure 7).

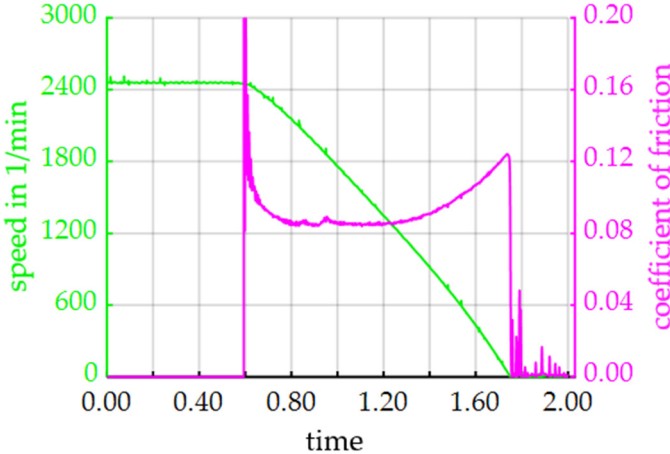

**Figure 7.** Friction coefficient curve for a shift at load level 6.

Figure 8a shows the steel and lining plates directly after the step test by pre-damaging the package. On the right side (Figure 8b). the condition of the plates after the continuous shifting test was documented. The photographs show that local discoloration and sinter transfer are no longer visible on the steel plates after the endurance test. The friction plate shows a stronger discoloration after the 10,000 shifts of the endurance test.

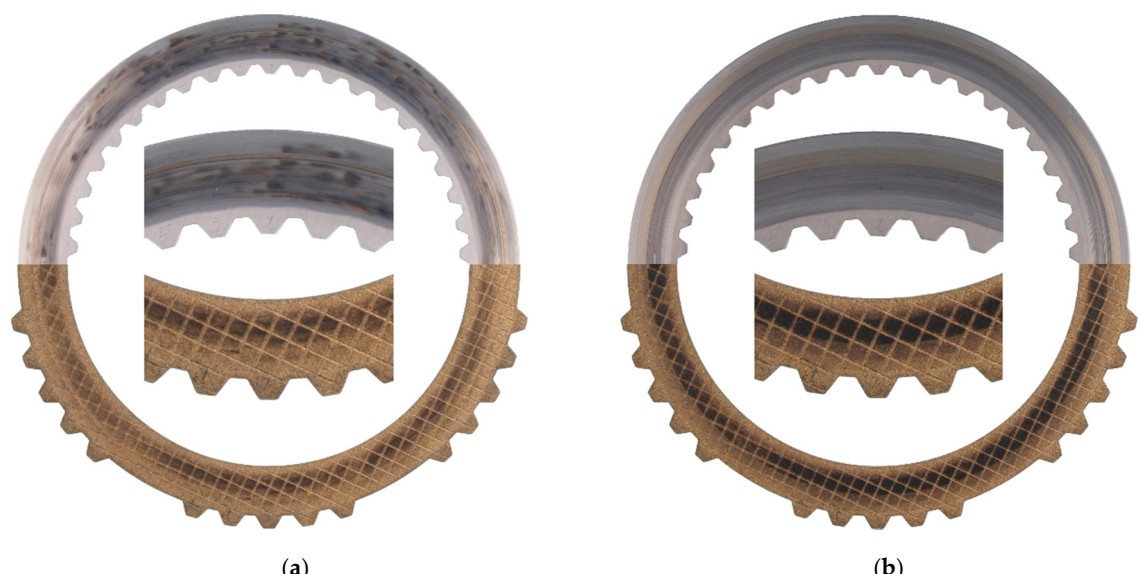

(**a**)　　　　　　　　　　　　　　　　　　　　　　　　(**b**)

**Figure 8.** (**a**) Photo of the pre-damaged plates after load level 6; (**b**) photo of the pre-damaged plates after the endurance test.

For test LL-08. the step test was run up to load level 8 (cf. Table 4). These loads resulted in local discoloration and a recognizable transfer of the sinter on the steel plate. In the friction coefficient curve of exemplary engagements at load level 8, a strong increase in the friction coefficient as a consequence of seizing can be seen (Figure 9). The step test was followed by the endurance test.

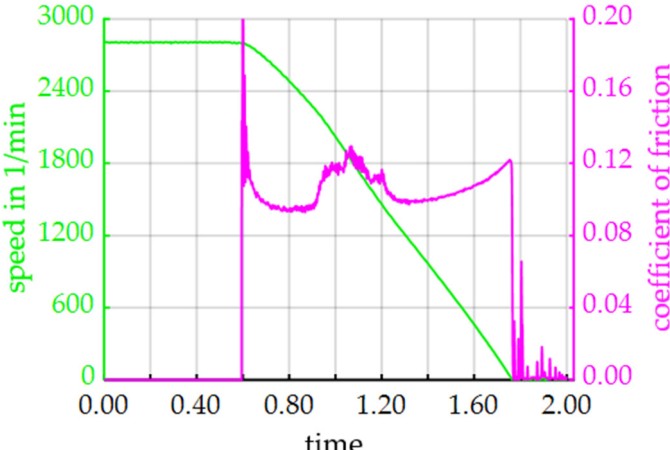

**Figure 9.** Friction coefficient curve for a shift at load level 8.

Figure 10a shows the steel and lining plates directly after the step test by pre-damaging the package. On the right side (Figure 10b). the condition of the plates after the continuous shifting test is documented. The photos show that local discoloration and sinter transfer are no longer visible on the steel plate after the endurance tests. In the areas of very significant discoloration on the steel plate prior to the endurance test, selective brightening can be seen after the endurance tests. The friction plate shows a stronger discoloration after the 10,000 shifts of the endurance test.

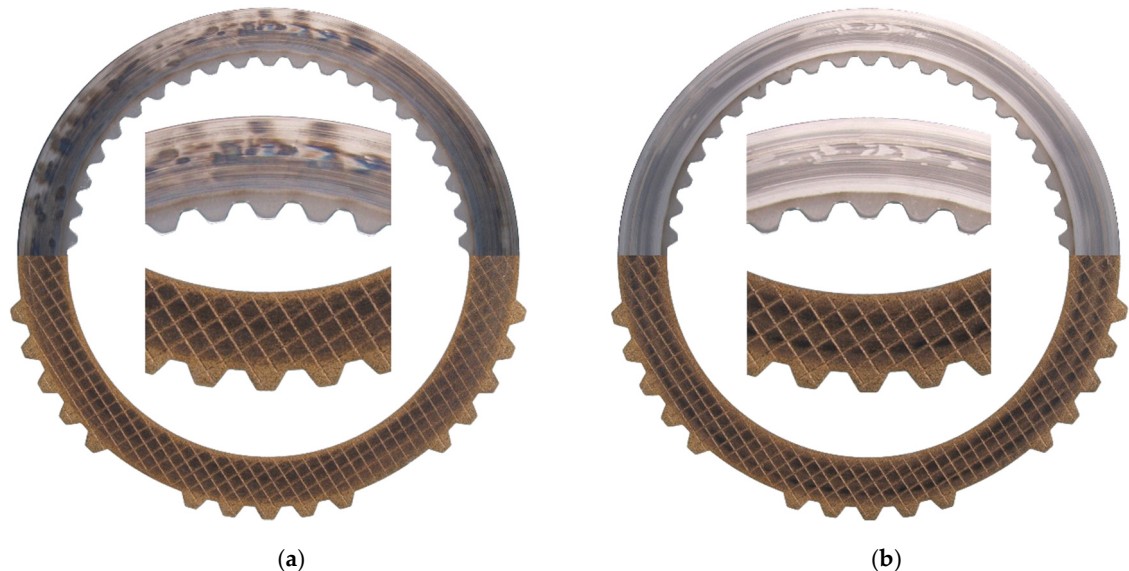

| (**a**) | (**b**) |

**Figure 10.** (**a**) Photo of the pre-damaged plates after load level 8; (**b**) photo of the pre-damaged plates after the endurance test.

Figure 11 shows the friction coefficient curves of the last 10 engagements and the first engagement during the running in for test LL-03. It can be seen that the friction coefficient curves of the last 10 engagements differ slightly compared to the first engagement. This knowledge is relevant for evaluating the following measurement records.

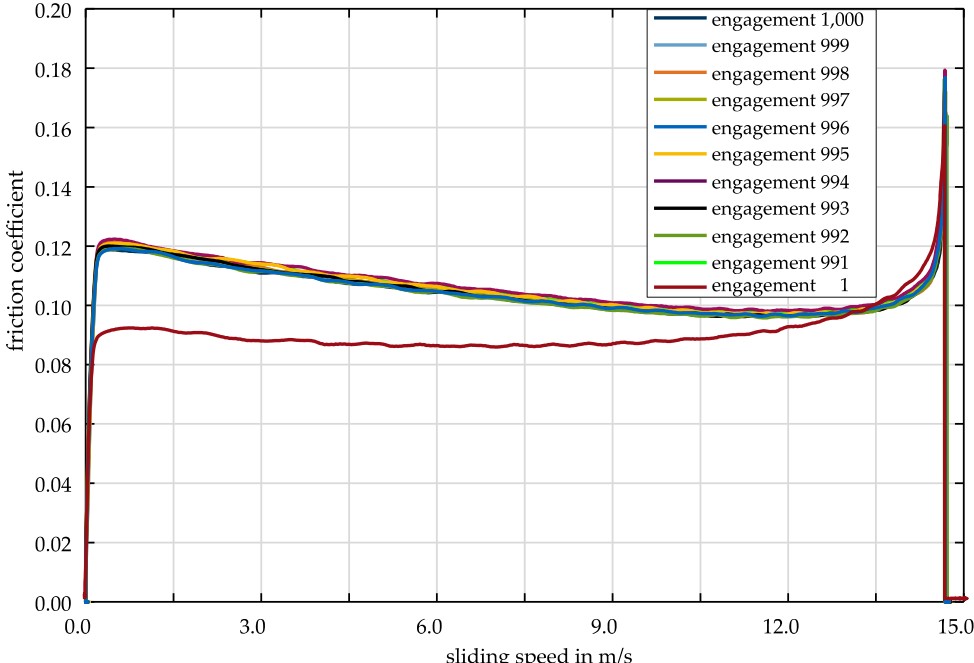

**Figure 11.** Friction coefficient curves of the last ten engagements in the run-in for test LL-03.

Figures 12–14 show the friction coefficient curves of three pre-damaged clutches (LL-03 to LL-08) and one reference clutch that was not pre-damaged (LL-00) for selected shifts in the endurance test. For shift 1 in the endurance test. the coefficients of friction over the entire speed range of the heavily pre-damaged clutches (pre-damaged at load levels 5 and 8) are slightly higher than with a slightly pre-damaged clutch (pre-damaged at load level 3) and a clutch that was not pre-damaged (cf. Figure 12).

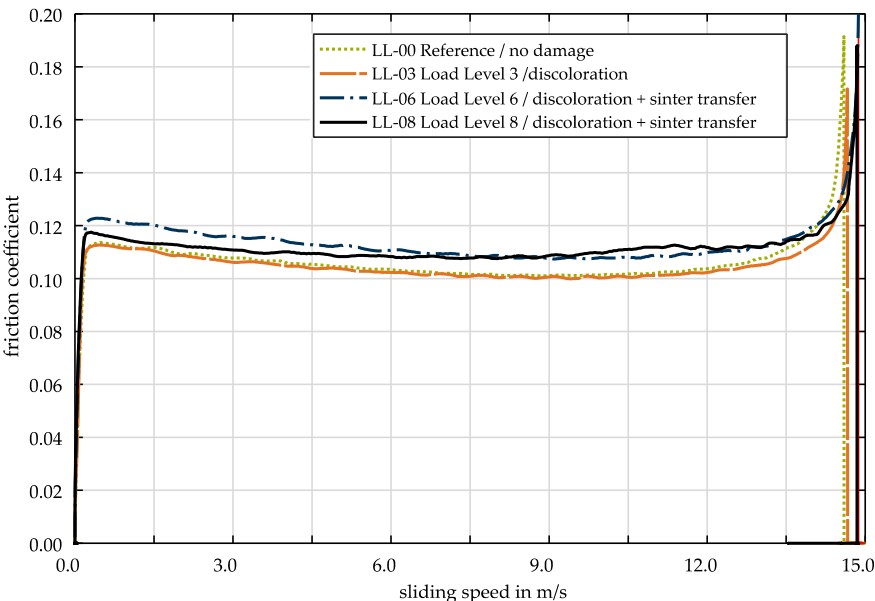

**Figure 12.** Single engagement comparison of previously damaged and non-damaged clutches (engagement 1 in the endurance test).

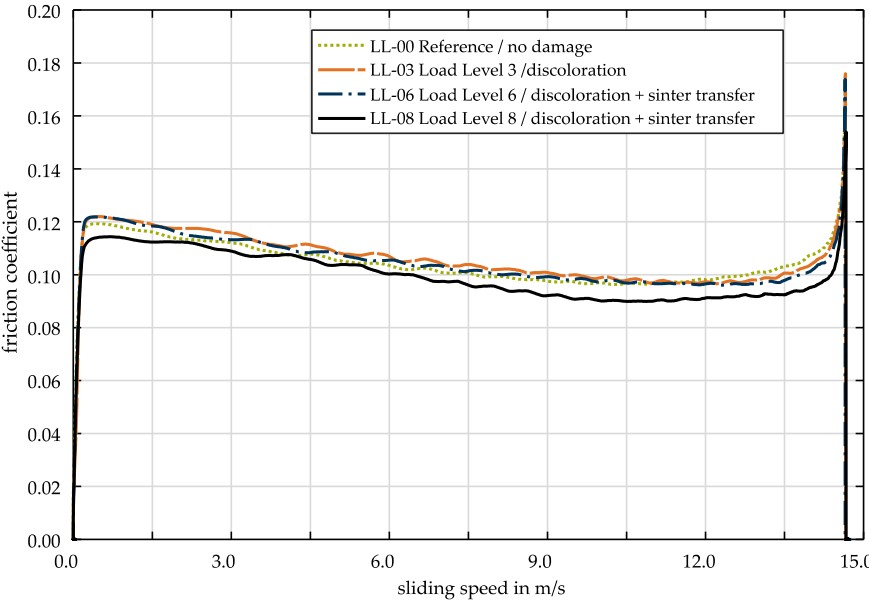

**Figure 13.** Single engagement comparison of previously damaged and non-damaged clutches (engagement 5000 in the endurance test).

Figure 13 shows values of the friction coefficients over the sliding speed for shift 5000 of the endurance tests. Clutch LL-08 shows a lower coefficient of friction level in comparison to the other clutches. In addition. the gradient of coefficient of friction over sliding speed with decreasing sliding speed is greater in the pre-damaged systems than in the reference clutch. This means that with decreasing sliding speed the coefficient of friction increases more strongly in the pre-damaged clutch than in the reference clutch. The speed-dependent fluctuations in the coefficient of friction of the pre-damaged clutches have increased in the endurance test compared with shift 1 (Figure 12).

For cycle 10,000 in the endurance test a similar picture as for shift 5000 appears (Figure 14). The increase in the coefficient of friction with a decrease of the sliding speed can be observed in the pre-damaged clutches. In addition, a stronger speed-dependent fluctuation of the coefficient of friction can be observed in the pre-damaged systems. The coefficient of friction level for clutch LL-08 is below

the level of the other clutches. This is strongly related to the fact that the friction coefficient level is very low in relative terms at the beginning of the shifting process. The coefficient of friction over sliding speed gradient behaves similarly to clutches LL-03 and LL-06.

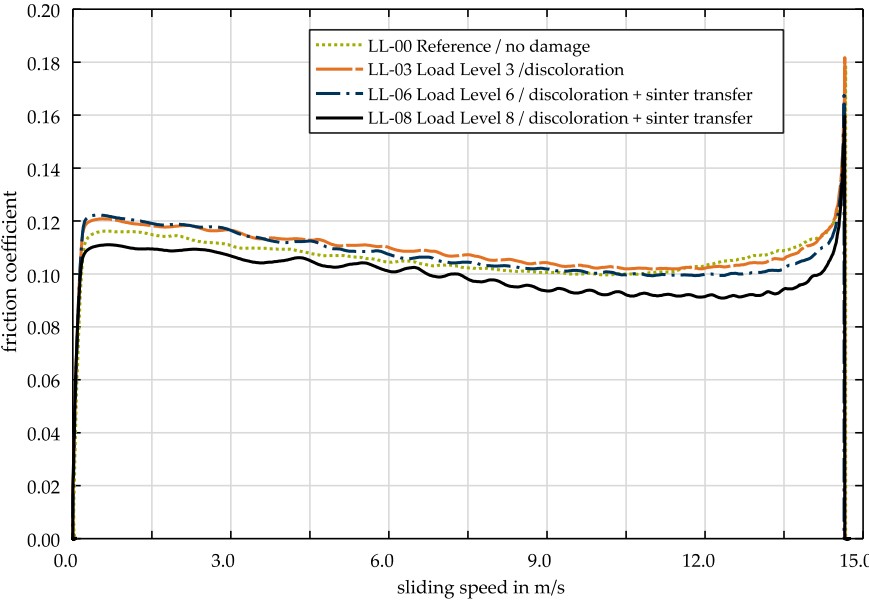

**Figure 14.** Single engagement comparison of previously damaged and non-damaged clutches (engagement 10,000 in the endurance test).

In addition to the individual shifts, Figure 15 shows the trend curves for $\mu_1$ of the four clutches investigated in the endurance test. The friction coefficient $\mu_1$ corresponds to the friction coefficient at 85% of the maximum sliding speed [26]. The trend curves show that $\mu_1$ in the reference clutch rises minimally during the endurance test. For clutches LL-03 and LL-06, $\mu_1$ decreases at the beginning and then rises slightly. LL-08 shows a continuous drop in the value $\mu_1$ over the duration of the endurance test and shows a clear difference to the other tests at the end.

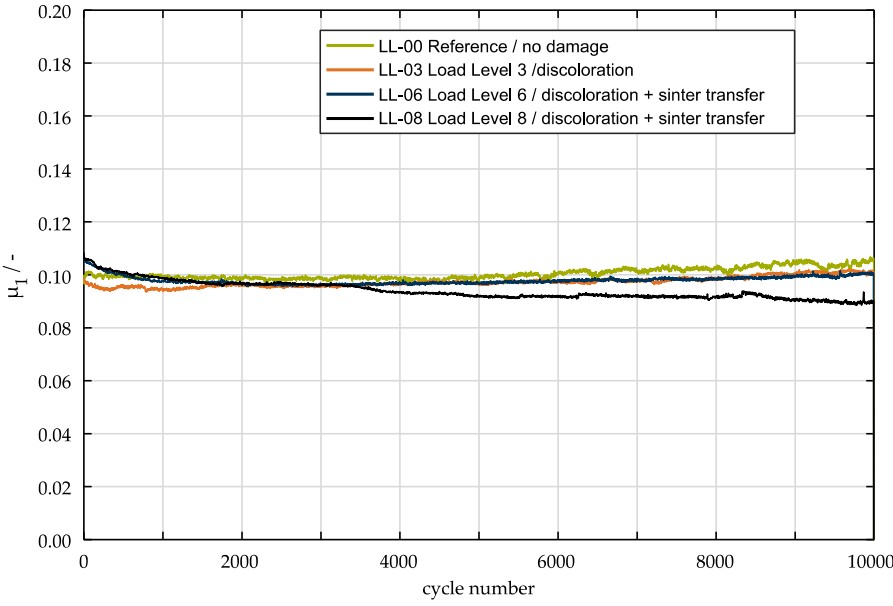

**Figure 15.** Trend comparison of $\mu_1$ over 10,000 shifts in the endurance test.

## 4. Discussions

In the step test. the different load levels cause varying forms of damage. Local discoloration, sinter transfer and friction coefficient fluctuations can be observed. After the endurance test, local discoloration and sinter transfer are no longer visible on the steel plate. This shows that this local discoloration, which is also described in the literature as oil-cracking products [13], is removed again due to wear mechanisms during the endurance test. It is also assumed that the sinter transferred is also removed due to wear during the endurance test. In the areas with very strong discoloration before the endurance test, specked brightening is visible after the endurance test. This brightening indicates structural changes as described by Kasem, Fairbank and Strebel [8,11,25]. These structural changes in the material cannot be visibly altered by wear mechanisms. A visible darkening of the lining plates in the continuous shifting test is caused by oil deposits and oil-cracking products in the pores. This was also observed by Acuner [26].

At the start of the endurance test. the values of coefficients of friction over the entire speed range of the heavily pre-damaged clutches are higher than with the slightly pre-damaged clutch and the clutch that had not been pre-damaged. Thus. the sinter transfer on the surface of the heavily damaged steel plate has an influence on the friction coefficient.

At the end of the endurance test it can be observed that with increasing pre-damage. the values of coefficients of friction at the beginning of the shift are lower. In the case of carbon synchronizer, this behavior was justified by the friction lining becoming smoother due to oil deposits [26]. It is assumed that this phenomenon also occurs here and that the pores are partially clogged up and smeared due to the damaging engagements.

## 5. Conclusions

In this paper the influence of spontaneous damage on the shifting behavior of clutches during extended use was investigated.

The clutches were pre-damaged in the step test, whereby the loads were increased in each load step. The load steps were determined from the friction work and friction power. Three clutches were pre-damaged with different intensities, so that spontaneous damage (local discoloration, sinter transfer, sinter transfer with increase of friction coefficient) occurred. This was followed by an endurance test (10,000 shifts) with the three pre-damaged clutches and a non-pre-damaged reference clutch. It should be noted that local discoloration and sinter transfer were no longer visible after the endurance tests. At the points of very pronounced discoloration before the continuous shifting test, spotty brightening was visible after the endurance test. At the start of the endurance test the values of coefficient of friction over the entire speed range of the heavily pre-damaged clutches were higher in comparison with the slightly pre-damaged clutch and the non-pre-damaged clutch. Sinter transfer on the surface of the heavily damaged steel plate, for this reason, has an influence on the coefficient of friction. At the end of the endurance test it could be observed that with increasing pre-damage the values of coefficient of friction increased more strongly with decreasing sliding speed. It is supposed that the friction lining became smoother due to oil deposits and that the pores were partially clogged and smeared by the damaging inserts. In addition, a correlation between the intensity of the pre-damage and the coefficient of friction at the start of a shift ($\mu_1$) was also shown. The greater the pre-damage. the lower was the coefficient of friction at the beginning of a cycle ($\mu_1$) at the end of the endurance test.

**Author Contributions:** Conceptualization, T.S.; Methodology, T.S.; Validation, T.S.; Formal analysis, T.S.; Investigation, T.S.; Resources, K.S.; Data curation, T.S.; Writing—original draft preparation, T.S.; Writing—review and editing, K.V., H.P. and K.S.; Visualization, T.S.; Supervision, H.P., K.V. and K.S.; Project administration, T.S. All authors have read and agreed to the published version of the manuscript.

**Funding:** The results presented are based on the research project FVA/FVV no. 515/V; self-financed by the Research Association for Drive Technology FVA (Forschungsvereinigung Antriebstechnik e.V.). The authors would like to express their thanks for the sponsorship and support received from the FVA and the members of the project committee. This work was supported by the German Research Foundation (DFG) and the Technical University of Munich (TUM) in the context of the Open Access Publishing Program.

**Conflicts of Interest:** The authors declare no conflicts of interest.

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
