# Peer review of "Friction Behavior of Pre-Damaged Wet-Running Multi-Plate Clutches in an Endurance Test"

_lubricants, doi:10.3390/lubricants8070068_

Round 1

Reviewer 1 Report

1. The parameters of the friction plate and the steel plate are not described, and you should list the parameters that have a greater impact on the friction behavior.

2. Three kinds of pre-damage of steel plate are studied in this paper. May I ask why you chose them? What are the causes of these injuries in practical applications?

3. We all know that friction plates will wear away the damage on the surface of steel plates over a long period of time. Therefore, what's the difference between this study and others? Have you done analysis and test on the impacts of the three kinds of pre-damages on important characteristics of steel plate, such as damage depth and damage area? Whether you should consider studying how long the pre-damage of steel plate will be worn away? How much does the thickness of the plate change over the test?

4.The conclusion should be modified illustrate the research results of the article more clearly.

Author Response

Point 1: The parameters of the friction plate and the steel plate are not described, and you should list the parameters that have a greater impact on the friction behavior.

Response 1: I added the parameters of the clutch in Table 1.

Point 2: Three kinds of pre-damage of steel plate are studied in this paper. May I ask why you chose them? What are the causes of these injuries in practical applications?

Response 2: When carrying out a step test until clutch failure due to the seizing of the plates, these three types of damage can be identified. The first form of damage on the steel plate is black discoloration. If the load is increased, sinter transfer occurs with slight changes in the friction coefficient. If the load is increased further, the friction coefficient fluctuations and the sinter transfer increase occurs due to seizure. In case the clutch fails, the sinter transfer respectively the seizure is so strong that the clutch is welded together. Three levels of damage were selected for these investigations, which are to be representative of the different levels of damage. In practical application, such damage occurs in cases of overload (e.g. emergency braking).

Point 3: We all know that friction plates will wear away the damage on the surface of steel plates over a long period of time. Therefore, what's the difference between this study and others? Have you done analysis and test on the impacts of the three kinds of pre-damages on important characteristics of steel plate, such as damage depth and damage area? Whether you should consider studying how long the pre-damage of steel plate will be worn away? How much does the thickness of the plate change over the test?

Response 3: In this paper the influence of spontaneous damage on the shifting behaviour of clutches during extended use is investigated. First of all, the clutches are subjected to such high loads that seizure occurs during the shifting process, which leads to a significant increase in the coefficient of friction and sinter transfer. It can be shown that even these extremely damaged clutches can still perform several thousand shifts in endurance tests. The friction behaviour of these clutches is compared and evaluated with less damaged clutches. In addition, it can be observed that the sinter transfer and local discoloration are worn away due to wear mechanisms. Due to very low wear rates of this tribo-system it is quite surprising that within several thousand of cycles recovery can be recognized already. Besides, not all effects of seizure at the friction interfaces may be repaired by wear; smoothening of surface structure and topography has a big impact on friction curve but may not be rebuilt by wear.

I updated the chapters “Material and Methods”, “Results” and “Conclusion” to make the research objective clearer.

Point 4: The conclusion should be modified illustrate the research results of the article more clearly.

Response 4: The conclusion was presented in more detail that the research findings have been better highlighted and become clearer.

Reviewer 2 Report

It is not clear to the reviewer why the authors refer to the step tests as a damaging procedure. In technical terms, being the clutch discs classified as wear components, one should refer to as pre-conditioning. Please, address this remark.

The authors should clarify how the specific friction work, expressed as [J/mm2] is computed. Is it normalised with respect to the clutch discs area?

At times the authors make use of unconventional unit of measurements when referring to the specific oil volume flow.

In Figure 3, it is not clear why the authors include both friction energy and power. Is the sliding time controlled? Being the friction power function of the time, what do the quasi-static values in Figure 3 represent?

In Figure 10, it would be much more interesting to compare the first and the last engagements. The reviewer does not notice great variability between the runs, which probably lies within the setup measurement accuracy. Please, verify that the measurement setup is capable of detecting such small friction variations.

Author Response

Point 1: It is not clear to the reviewer why the authors refer to the step tests as a damaging procedure. In technical terms, being the clutch discs classified as wear components, one should refer to as pre-conditioning. Please, address this remark.

Response 1: Right, for these investigations damaging procedure is part of the pre-conditioning for the investigations. In application, sinter transfer with friction coefficient peak up to 30% due to seizure is rather known as damage. I added this information to line number 140.

Point 2: The authors should clarify how the specific friction work, expressed as [J/mm2] is computed. Is it normalised with respect to the clutch discs area?

Response 2: Yes it is normalized with respect to the clutch area. I added this information to line number 133, 148, and 150.

Point 3: At times the authors make use of unconventional unit of measurements when referring to the specific oil volume flow.

Response 3: We used  mm3 /mm2 * s as a unit of measurement for the cooling oil volume flow. It describes the normalized volume of oil per second with respect to the clutch disc area and it is a very usual number for wet clutches. I added this information to line number 148. 

Point 4: In Figure 3, it is not clear why the authors include both friction energy and power. Is the sliding time controlled? Being the friction power function of the time, what do the quasi-static values in Figure 3 represent?

Response 4: We use here the maximum friction power at the beginning of the brake circuit. It is not the mean friction power. I added this information to line number 167.

Point 5: In Figure 10, it would be much more interesting to compare the first and the last engagements. The reviewer does not notice great variability between the runs, which probably lies within the setup measurement accuracy. Please, verify that the measurement setup is capable of detecting such small friction variations.

Response 5: I added the first engagement to the figure.